# Uses, Botanical Characteristics, and Phenological Development of Slender Nightshade (*Solanum nigrescens* Mart. and Gal.)

**DOI:** 10.3390/plants12081645

**Published:** 2023-04-14

**Authors:** Sara Monzerrat Ramírez-Olvera, Manuel Sandoval-Villa

**Affiliations:** Department of Soil Science, College of Postgraduate in Agricultural Sciences, Campus Montecillo, Montecillo 56264, Mexico; msandoval@colpos.mx

**Keywords:** Solanaceae, edible fruits and leaves, perennial plant, phenology

## Abstract

Slender nightshade (*Solanum nigrescens* Mart. and Gal.) is a perennial, herbaceous plant from the Solanaceae family, which is distributed in various environments. The aim of this study was to review the scientific literature and to establish slender nightshade plants under greenhouse conditions in order to record their phenological development. The specialized literature regarding the distribution, botanical characteristics, and uses of such species was analyzed. The phenological development was recorded based on the BBCH (Biologische Bundesanstalt, Bundessortenamt, Chemische Industrie) guide. Slender nightshade seeds were germinated under greenhouse conditions, then transferred to red porous volcano gravel locally known as tezontle in black polyethylene bags and watered with a Steiner nutrient solution. Changes in phenology were monitored and recorded from germination to the ripening of fruit and seeds. Slender nightshade has a wide distribution in Mexico and is used for medicinal and gastronomical purposes, as well as to control pathogens. The phenological development of slender nightshade has seven stages from germination to the ripening of fruit and seeds. Slender nightshade is a poorly studied plant with potential for human consumption. The phenological recording provides a tool for its management and further research as a crop.

## 1. Introduction

The topography of Mexico allows a heterogeneous climate that fosters the growth of a wide variety of plant species that are used as food, beverages, stimulants, medicine, and ornaments [1].

Mexico shelters 30,000 species, of which 7461 are useful plants and 2168 are edible [1]. However, some species are underutilized and considered as invasive weeds in conventional crop production [2].

Wild species of plants are sources of genes, metabolites, proteins, and enzymes with potential for human health; such features also help plants to resist diseases or adapt to changing climatic conditions [3]. Wild plants have high plasticity, which allows them to adapt to various environments, so their cultivation on a large scale could be considered, contributing to food security and diversification, since most of the world’s food intake depends on a group of plants that includes rice, corn, wheat, soybean, and potato [4].

The study of plant phenology allows for understanding the behavior of species during their life cycle, identifying quantitative changes in certain growth stages, and scheduling agricultural management practices, and is a useful tool for researchers, plant breeders, and producers [5].

Plant development is a process in which individual specimens or organs go through several identifiable stages during their life cycle, from cell differentiation to the emergence of organs and senescence [6]. The most relevant events in a plant’s life cycle can be assembled in a sequential manner on a certain scale [7].

The Biologische Bundesanstalt Bundessortenamt und Chemische Industrie (BBCH) scale is used to code phenologically similar growth stages of mono- and dicotyledonous plant species [8]. Currently, phenology scales have been proposed for various agricultural crops [9]. However, few studies are available for wild plants.

Slender nightshade (*Solanum nigrescens* Mart. and Gal.) is a wild species that is widely distributed in Mexico and used for medicinal, gastronomical, and pathogen-control purposes. However, questions about its uses exist, such as can it successfully complete its life cycle under controlled conditions in a greenhouse? Will the phenological record serve as a tool to determine the development and response of slender nightshade under controlled conditions from sowing to harvesting? Will it be possible to detect opportunities for the agronomical handling of slender nightshade? 

The objective of this study was to review the scientific literature on slender nightshade to establish plants under greenhouse conditions and record their phenological development. 

## 2. Results and Discussion

### 2.1. Slender and Divine Nightshade Plants

Slender nightshade (*Solanum nigrescens*) is a perennial, woody, and shrubby plant [10], whose common name is related to the European species (*Solanum nigrum* L.). In Mexico, it is also known as “*hierba de piojito*”, rabbit weed, and lion’s ear [11]. Although both species have very similar botanical characteristics [12] and are phenotypically very plastic [13], they have different life cycles, sizes, and leaf forms [12].

*S. nigrecens* is a perennial plant that grows up to 3.5 m tall [14], with grayish leaves up to 18 cm long and 10.5 cm wide [15] and with simple eglandular trichomes [10]. The inflorescences are clustered in cymes, with up to four with flowers per inflorescence [16], and the style is exserted [14]. *S. nigrum* is an annual plant [17], growing up to 80 cm tall [12,17], with leaves up to 7 cm long by 6 cm wide [12] and with multicellular and glandular trichomes [12]. The flowers are grouped into cymes of up to 10 flowers, and the styles do not extend beyond the anthers [13].

Although both species are sources of metabolites that confer medicinal properties on them [18,19], the fruit of *S. nigrum* contains nitrates and the steroid glycoalkaloids, α-solanine, and α-chaconine [20,21], associated with gastrointestinal and neurological disorders in human beings [22].

Moreover, the characteristics of the fruit and seeds make *S. nigrecens* easy for birds to distribute the plant [10]. It is considered a broadleaf weed in crops of agricultural interest, such as corn [11], sugar cane [23], and avocado [24].

#### 2.1.1. Distribution

Slender nightshade plants have high adaptability and tolerance to various environments, with morphological adaptations based on the environmental growth conditions [10].

Such plants are found in disturbed and non-disturbed environments, in warm and dry climates [10,25]. *S. nigrescens* is distributed at altitudes of 1500 to 3000 m in the southeast of the United States, Central America, the Caribbean, the Gulf of Mexico, Chile, and Argentina [10]. In Mexico, slender nightshade can be found in the states of Aguascalientes, Chihuahua, Durango, Jalisco, the State of Mexico, and Michoacan [15].

#### 2.1.2. Botanical Characteristics

*S. nigrescens* is an herbaceous plant (Figure 1a) that is erect, perennial, 1.5 to 3.5 m tall with branched stems, pubescent, and glabrous with age [14].

The leaf blades are lanceolate, 10 to 18 cm in length, 7 to 10.5 cm in thickness, acute or acuminate at the apex, and dentate at the edges, with petioles up to 4 cm in length (Figure 1b). The leaves are arranged in pairs, one larger than the other (Figure 1c) [14]. 

Flowers are grouped in racemes composed of four flowers (Figure 2a). The corolla is stellate (Figure 2b) with a diameter of 8–10 mm [16,26], with five white petals fused at the base and yellow and purple colors in the center. The stamens are yellow with anthers 2.5 to 3 mm in length [10], and the style is exserted (Figure 2c) [14]. The calyx is 1 to 3 mm in length [14], with five green sepals (Figure 2d).

The fruit clusters are composed of up to four fruit, with dropping pedicels that are weakly deflexed and 10 to 12 mm in length (Figure 3a) [10]. The fruit are globose, with a green (Figure 3b) to black color at ripening (Figure 3c) [14]. The calyx lobes are appressed, spreading at the fruit base [16]. The pulp is gelatinous and purple (Figure 3d). The seeds are lenticular in shape, 1 to 1.5 mm in diameter (Figure 3e), flattened, and brown in color, and the surfaces are minutely pitted [15].

#### 2.1.3. Uses

Slender nightshade has gastronomic, medicinal, insecticide, fungistatic, and bacteriostatic uses (Figure 4), depending on the useful part of the plant [27,28]. Its commercialization and sale in national markets is scarce [11]. Manual gathering is the main form of obtaining the plants.

##### Gastronomy

In Mexico, the stems and leaves are consumed as vegetables in soups, sauces, and broths or steamed [11,29]. In Tehuacan Puebla, it is one of the most important edible wild greens [30], with an average annual consumption rate of 8.2 kg per family [31], as well as being consumed in the Mixteca region [28].

Slender nightshade is a source of important trace elements for humans, with variations in the concentrations depending on the cooking method. For example, the boiled plant contains P, K, Ca, Mg, Na, Fe, Mn, Cu, and Zn at concentrations of 1850, 17,200, 33,500, 6000, 180, 240, 335, 10.1, and 55 mg per kg^−1^ of leaves, respectively. However, when cooked as a broth, the concentrations are 24, 495, 110, 39.8, 2.4, 0.2, 1.8, 0.1, and 0.6 mg per kg^−1^ of leaves, respectively [32].

##### Medicinal

In Mexico, slender nightshade is used to treat various illnesses. In the state of Hidalgo, the leaves are macerated for topical applications to reduce headaches, sore throats, and wounds [33]. In the state of Queretaro, the tender leaves are used to treat swollen feet [19].

In Guatemala, slender nightshade is useful against dermatological and gastrointestinal infections such as diarrhea, gastritis, stomach ulcers, and stomach discomfort [34]. In Ecuador, the leaves are applied topically or as an infusion to treat joint pain, headaches, and sore throats, as well as gastrointestinal, respiratory, and skin problems, such as bruises and inflammation [35,36], due to its effects on the digestive, nervous, and respiratory systems [37]. 

##### Biocidal Activity

Slender nightshade extract has been shown to inhibit the growth of pathogens due to the presence of metabolites such as coumarins, tannins, saponins, and anthraquinones [38], as well as cantalasaponin-3, a spirostanol glucoside [39].

Slender nightshade leaf extract inhibits the growth of *Candida albicans* [34,39,40], *Helicobacter pylori* [34], *Cryptococcus neoformans* [40], *Staphylococcus aureus*, *Microsporum canis*, *Microsporum gypseum*, *Trichophyton mentagrophytes,* and *Trichophyton rubrum* [34].

##### Insecticide

The aqueous extract of the entire slender nightshade plant at a 0.05% concentration causes 63% mortality in dengue-transmitting mosquito larvae (*Aedes aegypti*), while the root extract causes up to 98% mortality. The effect of the root extract can vary depending on the time of year, from 88 to 98% mortality. The month of April is the least effective and March the most effective for killing mosquito larvae [41].

### 2.2. Phenological Record

The description of the phenological growth stages of slender nightshade plants was carried out according to the scale of the BBCH phenological guide for plants of the Solanaceae family. The stages included were: germination (stage 0); leaf development (stage 1); formation of side shoots (stage 2); inflorescence emergence (stage 5); flowering (stage 6); development of fruit (stage 7); and ripening of fruit and seeds (stage 8). The stem elongation stage (stage 3), which coincides with leaf development, was omitted. In the development of harvestable vegetative parts (stage 4), the fruit were considered as harvestable organs. Senescence (stage 9) was included because slender nightshade is a perennial plant.

#### 2.2.1. Main Stage 0: Germination

This stage describes the period from the sowing of the dry seed to the emergence of the seedling when the cotyledons break through the soil surface (Table 1). It includes the secondary stages (Figure 5), seed imbibition (BBCH codes 01 and 03), radicle emergence (code 05), breaking of the seed coat (code 07), and emergence of the seedling (code 09). Under the conditions evaluated in this research, germination occurred in 14 days.

The hardness of the seed testa in slender nightshade seeds is a physical barrier to imbibition and delays germination. In wild ecosystems, this is an important feature [42], given that if germination occurs at an inappropriate time, the survival of the species will be compromised [43]. In nature, seed latency can be interrupted by fluctuating (freezing–thawing) temperatures and the passage through the digestive tract of animals [44]. Since physical latency is not a desirable trait in production, it should be a feature to consider in the domestication process of slender nightshade.

#### 2.2.2. Principal Stage 1: Leaf Development

The leaf development phenological stage (Table 2) includes the unfolding of the opposite cotyledons until the emergence and unfolding of new true leaves on the main shoot (Figure 6a).

The true leaves and shoots are pubescent (Figure 6b), which is considered a strategy for tolerating various types of stress [45], such as UV damage, water loss, and herbivory [46]. Densely pubescent leaves are common in plants grown in environments with low precipitation [47].

The first three leaves of slender nightshade are oval with smooth edges, afterwards turning serrated and pointed (Figure 6c). Leaf serration is one of the morphological changes between juvenile and adult leaves [48]. The leaves become more serrated as the plant ages [49]. Other changes include the relationship between the length and width of the leaves [50,51].

During the leaf development stage, the cotyledons gradually turn yellow (Figure 6d), the height of the seedling and the root length increase, and the development of the roots is greater than the shoots (Figure 6a). Favoring root growth rather than shoot growth facilitates the acquisition of water and nutrients [52] and enables growth in competitive environments, as well as tolerance to water scarcity [53]. These features may change during domestication [54].

Slender nightshade plants have a high capacity for adaptation and tolerance to various environments due to morphological adaptations [10].

#### 2.2.3. Principal Stage 2: Formation of Side Shoots

The first primary apical side shoot becomes visible, followed by the second side shoot, until the appearance of nine or more side shoots (Table 3).

During this stage, the plants increase in height, cotyledons abscise, and the leaves gradually change from an intense green to a grayish-green coloration (Figure 7). Under the conditions evaluated in this research, the emergence of side shoots occurred when the plants reached 5 cm in height and unfolded the sixth true leaf.

In natural conditions, slender nightshade grows in competition with other plant species. The formation of shoots allows for lateral growth, related to increased light capture and shading of neighboring species [55,56].

Excessive foliage will be a feature to consider in the agronomic management of fruit production. The use of pruning schedules will allow for a balance between the partitioning of photoassimilates between source and sink organs [57].

#### 2.2.4. Principal Stage 5: Inflorescence Emergence

The first inflorescence bud is visible until nine or more inflorescences emerge (Table 4). This stage begins with the emergence of the flower bud and ends with the presence of the first petal, when the flowers are still closed (Figure 8a).

The first flower bud appears after the bifurcation of the main shoot (Figure 8b), followed by the emergence of the eleventh bud. This stage includes the emergence and development of the flower buds, the enlargement of the inflorescences, and the anthesis of the flowers.

At emergence, the flower buds are erect and merged, with closed sepals (Figure 8c), then they are free and erect (Figure 8d), at level with the leaves (Figure 8e), with 15% of the sepals open (Figure 8f), 70% of the sepals open, and 60% of the petals visible (Figure 8g), with the sepals completely opened and with the petals completely visible (Figure 8h). 

The sepals and petals of the flower buds are densely pubescent (Figure 9a–c). Pubescence of the floral organs is related to the regulation of temperature and light intensity [58], representing a strategy to ensure reproductive success by maintaining a microclimate within the flower buds [59].

#### 2.2.5. Principal Stage 6: Flowering

Flowering starts when the first flower of the first inflorescence opens and it ends when more than nine flowers of the ninth inflorescence open (Table 5; Figure 10a). 

The opening of flowers in the racemes is gradual, beginning with 25 (Figure 10b), 50 (Figure 10c), and 75% (Figure 10d) of flowers open in the raceme. Afterwards, 25, 50 (Figure 10e), and 75% (Figure 10f) of the petals wilt and the fruit set in the inflorescence becomes visible (Figure 10g). Flowering ends with the wilting and dehiscence of the petals in most of the inflorescences.

During flower opening (Figure 11a), stamens are exposed to attract pollinators [60]. After pollination, petals complete their function and senescence of the flowers begins. The flower gradually closes (Figure 11b,c) and the petals wilt, leading to a complete change in the perianth coloration. The removal of sink tissues such as petals and anthers allows for the mobilization and recycling of macromolecules and nutrients for the development of embryos and other organs [61].

#### 2.2.6. Principal Stage 7: Development of Fruit

Stage seven begins when the first fruit of the first fruit cluster reaches the typical size (Table 6); subsequently, the second cluster reaches the typical size, and the process continues until the first fruit of nine or more clusters reach the typical size.

This stage of fruit development includes green fruit surrounded by sepals and the total growth of the fruit (Figure 12a–d). The increase in fruit size is due to cell division and expansion. Under the conditions evaluated in this study, the fruit reached up to 1.5 cm in diameter.

The growth of the fruit within a fruit cluster is not uniform. In the early stages, 50% of the fruit have reached 15% of their final size, with senescent petals attached, whereas the other 50% of the fruit have reached 25% of their final size (Figure 13a). The dry petals fall off and the fruit surrounded by sepals continues to develop until 50% of the fruit reach around 40% of their final size and 50% of the fruit reach around 25% of their final size (Figure 13b), after which the fruit continue to grow (Figure 13c).

#### 2.2.7. Principal Stage 8: Ripening of Fruit and Seed

Fruit ripening starts when the green fruit gradually turn purple (Table 7). In this study, the ripening of all fruit within a cluster happened 143 days after the appearance of the flower bud.

The fruit continue to expand and a purple pigmentation begins to appear (Figure 14a), which increases (Figure 14b) until 50% of the fruit’s surface turns purple (Figure 14c). Finally, 100% of the fruit become pigmented (Figure 14d), then the intensity of the green color of the sepals diminishes (Figure 14e) and turns brown (Figure 14f). During ripening, the fruit show a slow growth and have intense metabolic changes [62].

Slender nightshade fruit are vital organs for preserving the species. They are confirmed by the receptacle of the seed developed from a mature and fertilized ovary. The concentration of phytohormones in the seed is correlated with the size and shape of the fruit [62].

The purple pigmentation of slender nightshade fruit is related to the presence of anthocyanins, which are useful for the plant to tolerate various biotic and abiotic stresses; such pigments are also important for human health [63], since they are used for treating diseases such as cancer [64], due to their antioxidant capacity and their role in removing reactive oxygen species [65].

## 3. Materials and Methods

### 3.1. Review of the Scientific Literature

A literature search was carried out in the Bielefeld Academia Search Engine (BASE), the Biodiversity Heritage Library (BHL), Connecting Repositories Aggregating the World (CORE), Dialnet, Europe PMC, Google Scholar, the International Information System for Agricultural Science and Technology (AGRIS), Open Access, Plantnet, Portal Network Botanical Specimens (SEINet), PubMed research papers, Scopus, Science Direct, Science Research, and World Wide Science. 

The search included the keywords “coastal-dune nightshade”, “slender nightshade” “divine nightshade”, “chiquiquelite”, “chichaquelite”, “hierba mora,” “mambia”, “maniloche”, “macuy”, and “*Solanum nigrescens*” in order to identify scientific reports related to the botanical characteristics; distribution; traditional, medicinal, and culinary uses; and bioassays that have used the plant for pathogen control. 

### 3.2. Phenological Recording

Wild slender nightshade plants were gathered in the “Siberia” plot of land located in San Luis Huexotla, Texcoco, in the State of Mexico, at 19°28′14″ N and 98°51′21″ W, at a height of 2330 m during October of 2021. The average maximum and minimum temperatures in the aforesaid area during October were 23.6 °C and 10.9 °C, respectively. The temperature data were obtained from the agrometeorological station in Montecillo, Texcoco, in the State of Mexico.

The biggest plant was chosen, with ripe fruit and being free of plagues and other diseases. Ten large, fully developed, purple-colored fruit were gathered. The fruit were rubbed against each other and the outer skins and seed pulp were separated and rinsed in distilled water until the pulp had been fully removed. The seeds were then extracted and dried at room temperature (24 °C).

The mature fruit of slender nightshade were washed to remove the pulp, then the seeds were extracted and dried at room temperature. Afterwards, the seeds were placed in 200-cavity expanded polystyrene trays in a peat substrate. At 44 days after planting, the seedlings were transferred to black polyethylene bags containing tezontle (red porous volcano gravel) as the substrate. Their nutrition was based on the Universal Steiner nutrient solution [66]. The plants were established under greenhouse conditions with UVII-720 plastic, 85% light transmission, and respective average temperatures of 27 °C and 16 °C during the day and dark hours, respectively. The relative humidity was 55%. When the plants had mature fruit, a plant with flowers and fruit was selected and deposited in the CHAPA Hortorial Herbarium with voucher number 155812. 

The phenological changes were recorded from germination, leaf development, formation of the side shoots, inflorescence emergence, flowering, fruit development, and ripening of the fruit and seeds. The description included photographs of the most relevant stages. The photographs were taken during the period of February to September of 2022, with the super-macro function of the SM-A022M camera at 2.0 MP, fitted with an f/1.9 lens of an A02 Samsung Galaxy mobile phone A02 (Android version 10.0). 

The structures of the buds and flowers were photographed using a Canon DS126621 camera (Canon, Tokyo, Japan) connected to a SteREO Discovery V20 stereo microscope (Zeiss, Jena, Germany).

The photographs were edited using Gimp 2.10 software (www.gimp.org, accessed on 18 Octuber 2022), and the scale was indicated using ImageJ software [67].

## Figures and Tables

**Figure 1 plants-12-01645-f001:**
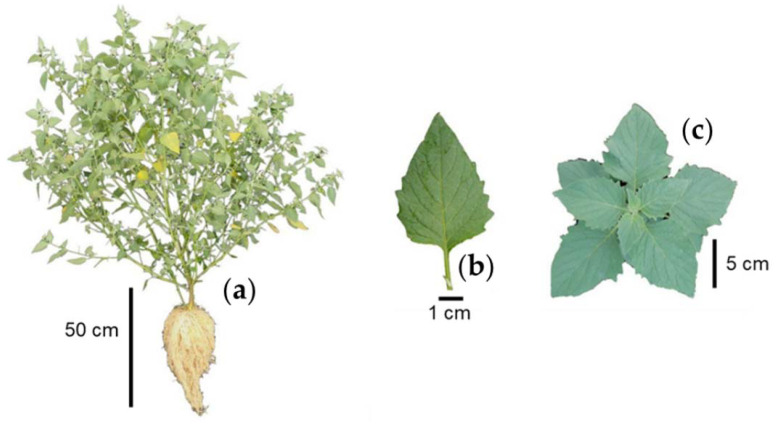
Whole plant (**a**), characteristics (**b**), and arrangement (**c**) of leaves of slender nightshade plant (*Solanum nigrescens*). Photos by Sara Monzerrat Ramírez-Olvera.

**Figure 2 plants-12-01645-f002:**
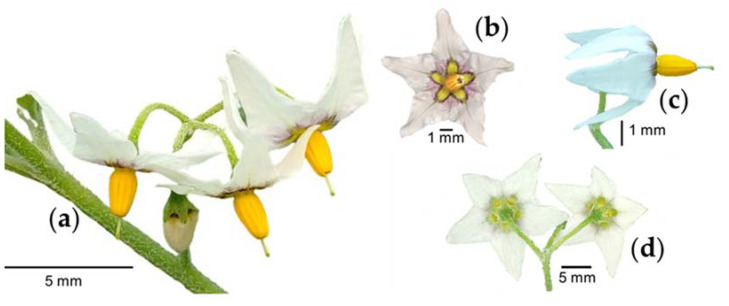
Flowers grouped in racemes (**a**), corolla stellate (**b**), style exserted (**c**), and calyx (**d**) of slender nightshade plant (*Solanum nigrescens*). Photos by Jorge Manuel Valdez-Carrasco (**b**) and Sara Monzerrat Ramírez-Olvera (**a**,**c**,**d**).

**Figure 3 plants-12-01645-f003:**
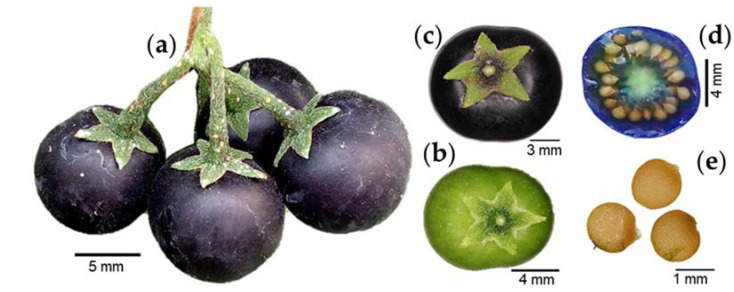
Fruit clusters (**a**), immature (**b**) and ripe fruit (**c**), pulp (**d**), and seeds (**e**) of slender nightshade plant (*Solanum nigrescens*). Photos by Sara Monzerrat Ramírez-Olvera (**a**,**d**) and Jorge Manuel Valdez-Carrasco (**b**,**c**,**e**).

**Figure 4 plants-12-01645-f004:**
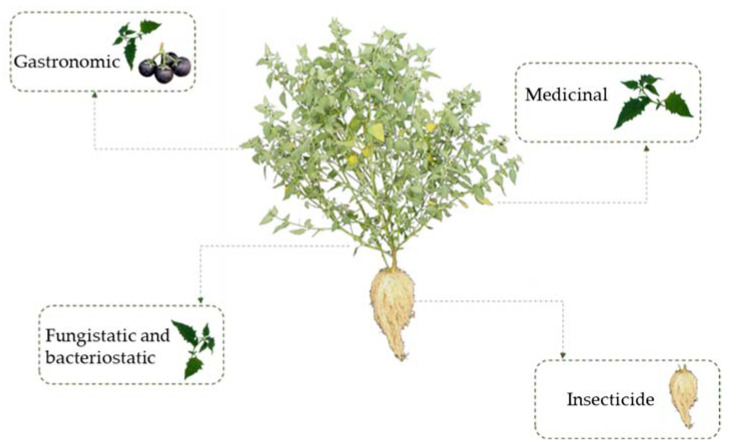
Main uses of slender nightshade plant (*Solanum nigrescens*). Photos by Sara Monzerrat Ramírez-Olvera.

**Figure 5 plants-12-01645-f005:**
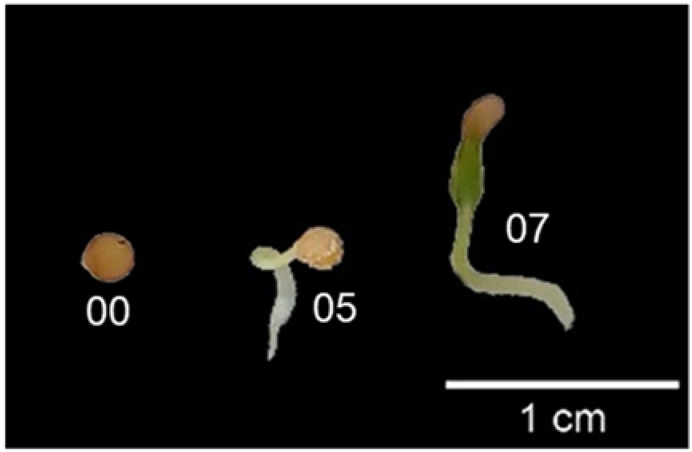
Main stage 0 (germination) for slender nightshade seedlings (*Solanum nigrescens* Mart. and Gal.), based on the BBCH scale: 00: dry seed; 05: radicle emerged from seed; 07: hypocotyl with cotyledons breaking through seed coat. Photos by Sara Monzerrat Ramírez-Olvera.

**Figure 6 plants-12-01645-f006:**
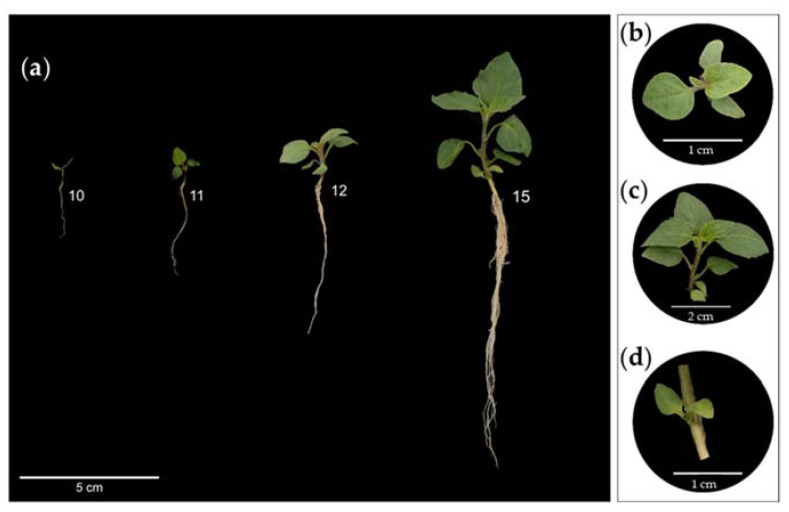
Principal stage 1 (leaf development) (**a**), pubescent (**b**) and serrated (**c**) leaves, and cotyledons (**d**) of slender nightshade plants (*Solanum nigrescens*): 10: cotyledons completely unfolded; 11: first true leaf on main shoot fully unfolded; 12: second leaf on main shoot fully unfolded; 15: fifth leaf on main shoot fully unfolded. Photos by Sara Monzerrat Ramírez-Olvera.

**Figure 7 plants-12-01645-f007:**
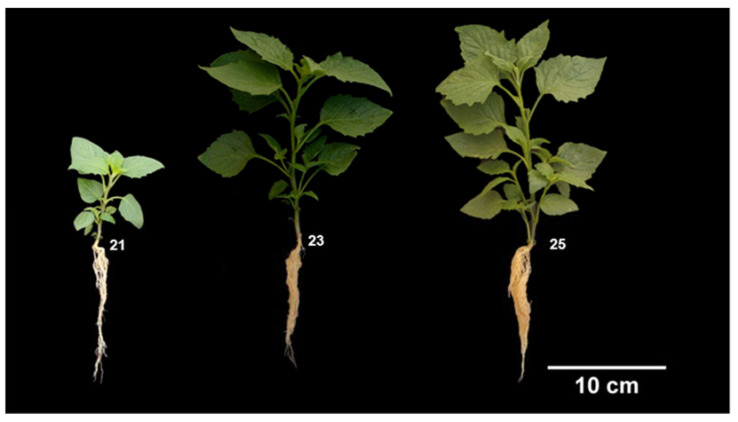
Principal stage 2 (formation of side shoots) of slender nightshade plants (*Solanum nigrescens*) based on the BBCH scale: 21: first primary apical side shoot visible; 23: third primary apical side shoot visible; 25: fifth primary apical side shoot visible. Photos by Sara Monzerrat Ramírez-Olvera.

**Figure 8 plants-12-01645-f008:**
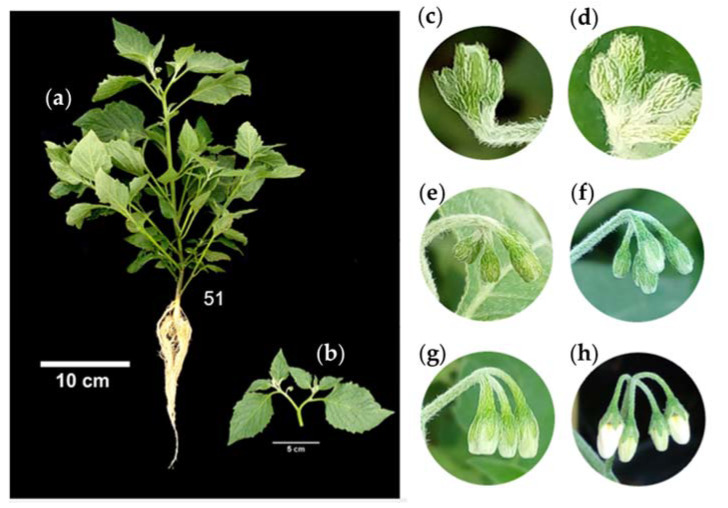
Inflorescence emergence (**a**), bifurcation of the main shoot (**b**), erect flower buds (**c**), and free buds (**d**) at level with the leaves (**e**), with 40% of the petals visible (**f**), 70% of the petals visible (**g**), and the sepals completely opened and with petals completely visible (**h**) in slender nightshade plants (*Solanum nigrescens*). Photos by Sara Monzerrat Ramírez-Olvera.

**Figure 9 plants-12-01645-f009:**
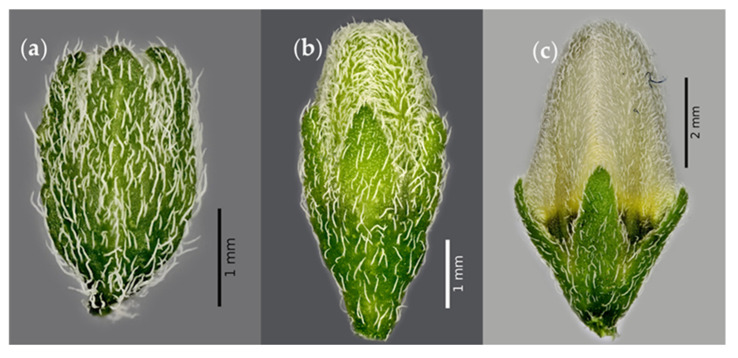
Flower buds of slender nightshade plants (*Solanum nigrescens*) at 5 (**a**), 20 (**b**), and 100% (**c**) open. Photos by Jorge Manuel Valdez-Carrasco.

**Figure 10 plants-12-01645-f010:**
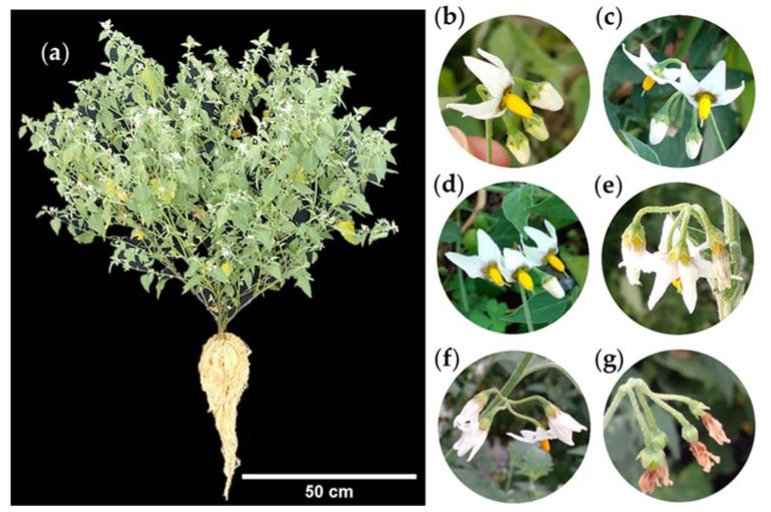
Flowering (**a**) of slender nightshade plants (*Solanum nigrescens*); inflorescence with 25 (**b**), 50 (**c**), and 75% (**d**) fertilized blossoms (**e**) developing fruit (**f**); and fruit set (**g**). Photos by Sara Monzerrat Ramírez-Olvera.

**Figure 11 plants-12-01645-f011:**
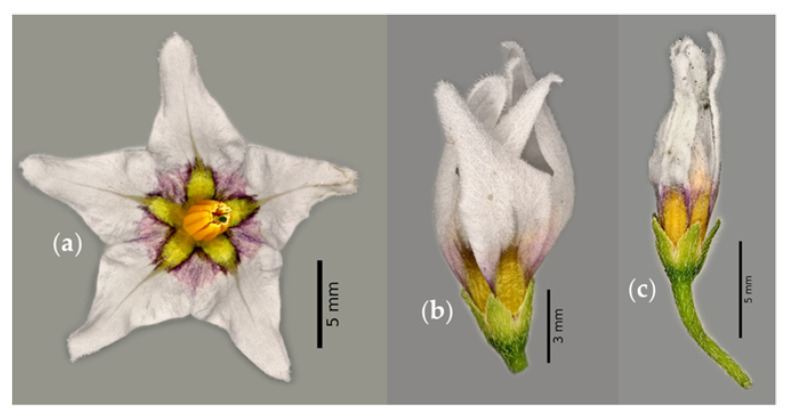
Flower of slender nightshade plant (*Solanum nigrescens*) open (**a**) and with 80 (**b**) and 90% closed flowers (**c**). Photos by Jorge Manuel Valdez-Carrasco.

**Figure 12 plants-12-01645-f012:**
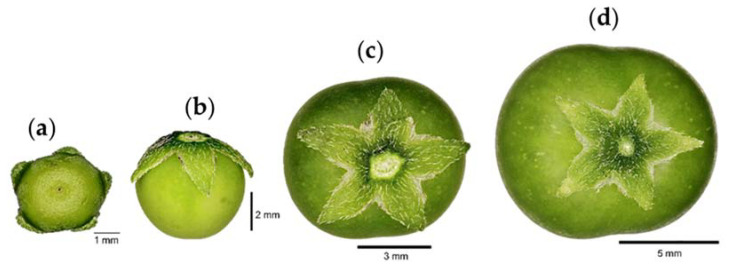
Fruit of slender nightshade plants (*Solanum nigrescens*) with the 5 (**a**), 20 (**b**), 70 (**c**), and 100% (**d**) of their final size. Photos by Jorge Manuel Valdez-Carrasco.

**Figure 13 plants-12-01645-f013:**
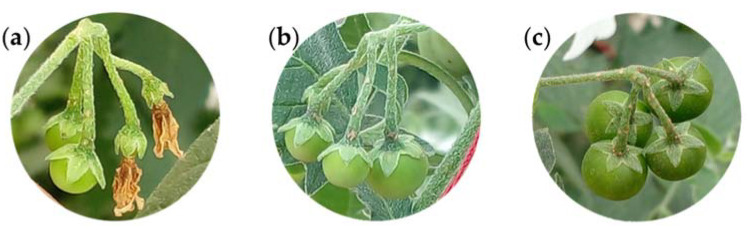
The 50% of fruit of slender nightshade plants (*Solanum nigrescens*) have reached 15 (**a**), 30 (**b**), and 80% (**c**) of their final size. Photos by Sara Monzerrat Ramírez-Olvera.

**Figure 14 plants-12-01645-f014:**
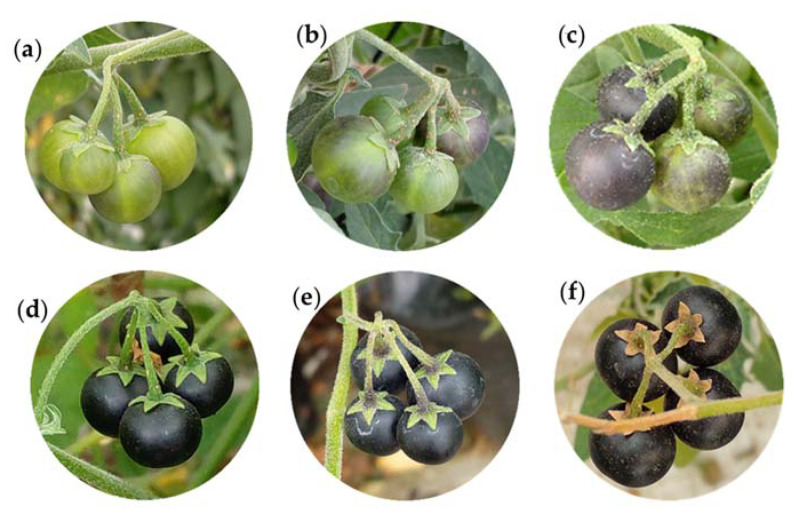
Fruit cluster of slender nightshade plants (*Solanum nigrescens*) with 10 (**a**), 30 (**b**), 50 (**c**), and 100% (**d**) purple pigmentation on the surface. The sepals are green-yellow (**e**) and withered (**f**). Photos by Sara Monzerrat Ramírez-Olvera.

**Table 1 plants-12-01645-t001:** Description of main stage 0 (germination) for slender nightshade plants, based on the BBCH scale.

Code	Description
00	Dry seed (Figure 5).
01	Beginning of seed imbibition.
03	Seed imbibition complete.
05	Radicle emerged from seed (Figure 5).
07	Hypocotyl with cotyledons breaking through seed coat (Figure 5).
09	Emergence: Cotyledons break through soil surface.

**Table 2 plants-12-01645-t002:** Description of principal stage 1 (leaf development) of slender nightshade, based on the BBCH scale.

Code	Description
10	Cotyledons completely unfolded (Figure 6a).
11	First true leaf on main shoot fully unfolded (Figure 6a).
12	2nd leaf on main shoot fully unfolded (Figure 6a).
13	3rd leaf on main shoot fully unfolded.
14	4th leaf on main shoot fully unfolded.
15	5th leaf on main shoot fully unfolded (Figure 6a).
16	6th leaf on main shoot fully unfolded.
1.	Stages continuous until…
	9 or more leaves on main shoot unfolded.

**Table 3 plants-12-01645-t003:** Description of the principal stage 2 (formation of side shoots) of slender nightshade, based on the BBCH scale.

Code	Description
21	First primary apical side shoot visible (Figure 7).
22	2nd primary apical side shoot visible.
23	3rd primary apical side shoot visible (Figure 7).
24	4th primary apical side shoot visible.
25	5th primary apical side shoot visible (Figure 7).
27	7th primary apical side shoot visible.
29	9 or more apical primary side shoots visible.

**Table 4 plants-12-01645-t004:** Description of principal stage 5 (inflorescence emergence) of slender nightshade, based on the BBCH scale.

Code	Description
51	First inflorescence visible (first bud erect).
52	2nd inflorescence visible (first bud erect).
53	3rd inflorescence visible (first bud erect).
55	5th inflorescence visible.
57	7th inflorescence visible.
59	9 or more inflorescences visible.

**Table 5 plants-12-01645-t005:** Description of the principal stage 6 (flowering) of slender nightshade, based on the BBCH scale.

Code	Description
61	First inflorescence with first flower open.
62	2nd inflorescence with first flower open.
63	3rd inflorescence with first flower open.
65	5th inflorescence with first flower open.
67	7th inflorescence with first flower open.
69	9 or more inflorescences with open flowers.

**Table 6 plants-12-01645-t006:** Description of principal stage 7 (development of fruit) of slender nightshade, based on the BBCH scale.

Code	Description
71	First fruit cluster, first fruit has reached typical size.
72	2nd fruit cluster, first fruit has reached typical size.
73	3rd fruit cluster, first fruit has reached typical size.
75	5th fruit cluster, first fruit has reached typical size.
77	7th fruit cluster, first fruit has reached typical size.
79	9 or more fruit clusters with fruit of typical size.

**Table 7 plants-12-01645-t007:** Description of the principal stage 8 (ripening of fruit and seeds) of slender nightshade, based on the BBCH scale.

Code	Description
81	10% of fruit show typical fully ripe color.
82	20% of fruit show typical fully ripe color.
83	30% of fruit show typical fully ripe color.
84	40% of fruit show typical fully ripe color.
85	50% of fruit show typical fully ripe color.
86	60% of fruit show typical fully ripe color.
87	70% of fruit show typical fully ripe color.
88	80% of fruit show typical fully ripe color.
89	Fully ripe: fruit have typical fully ripe color.

## Data Availability

The available data are presented in the manuscript.

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
