# Peer review of "Uses, Botanical Characteristics, and Phenological Development of Slender Nightshade (Solanum nigrescens Mart. and Gal.)"

_plants, 2023, doi:10.3390/plants12081645_

Round 1
Reviewer 1 Report
The manuscript is nice and to the point. Strength is also a skillful use of drawings. After making corrections, it is suitable for publication in the journal Plants.
a) At the end of the introduction, please give list the research questions you raised before starting the planting experiments.
b) In the Results and discussion chapter, mention only what results you got and what you think about them. Thus, the whole of 2.1 to 2.1.3.4. is actually part of the material and methodology. You describe the plant you are studying through other researchers. So this is the background knowledge that goes to the material chapter.
c) Line 345-346: "The mature fruits of divine nightshade were washed to remove the pulp, the seeds were extracted and dried at room temperature.". About what was the room temperature at that time? Describe in more detail the method of seed collection. When (date and year) were they collected? Was it a wild plant from which the seeds were collected? Were seeds collected from only one plant? Describe the place where the plant grows and give the region where it grew (coordinates or the exact village and county)? Has the author observed variability in wild species (e.g. were the seeds of the plant with the largest fruit collected)? Was a herbarium specimen also collected from the plant (if so, where herbarium specimen is it stored now?) or only the fruits? Are the seeds left over from the experiment stored somewhere, if so, in which institution and what is the deposit number?
d) Line 340-341: „The search included the keywords "divine nightshade," "hierba mora," "macuy," and "Solanum nigrescens" /---/“. But this plant has at least 26 Latin synonyms: https://powo.science.kew.org/taxon/urn:lsid:ipni.org:names:239356-2#synonyms In addition, in the English-language literature, this plant is called „Divine nightshade, Coastal-dune Nightshade, Slender nightshade“: https://identify.plantnet.org/the-plant-list/species/Solanum%20nigrescens%20M.%20Martens%20%26%20Galeotti/data . In Spanish, this plant is called "chiquiquelite, chichaquelite, mambia, hierba mora, maniloche, pitoxe": https://swbiodiversity.org/seinet/taxa/index.php?tid=3839&taxauthid=1&clid=2765# So the authors should better justify why they chose only a limited number of English and Spanish plant names for the search? They should also justify why Latin synonyms were not searched for when searching for information about the plant.
e) The authors have thoroughly searched informations in several databases. Just for the authors to know in the future that older literature can also be searched in such databases: https://www.biodiversitylibrary.org/ ; https://bibdigital.rjb.csic.es/en/ . However, the information contained in the books can be searched in this database: https://books.google.com/ . The reviewer saw that there is quite a lot of information about this plant in the books.google.com databases. Of course, the authors have already covered most of the data on this topic in the manuscript. Therefore, it is not necessary to perform a new search in the databases. However, I also recommend using the book "Ethnobotany of Mexico" (2016) (pages 186 and 301 talk about this plant), which has not been used by the authors.
f) Figure 10. The terms in the explanation of the figure should be changed. For example, instead of "open flowers" you could use "full-blown"; "fertilized blossom" could be used instead of "dead flowers"; instead of "fruit set" it could be either "developing fruits" or "ripening fruits".
g) Line 355. “The description included photographs of the most relevant stages.” Please write who took the plant photos in this manuscript? What month and year were the photos taken? Were the photos also processed or manipulated somehow (with what program?)? Please also indicate which camera the photos were taken with.
Author Response
Response to Reviewer 1
a) At the end of the introduction, please give list the research questions you raised before starting the planting experiments.
Response: The following research questions were included:
What is currently known about the uses and characteristics of slender nightshade? Can slender nightshade successfully complete its life cycle under controlled greenhouse condition? Will the phenological record serve as a tool to determine the development of slender nightshade from sowing to harvesting? Can opportunities be found for the agronomic management of slender nightshade?
b) In the Results and discussion chapter, mention only what results you got and what you think about them. Thus, the whole of 2.1 to 2.1.3.4. is actually part of the material and methodology. You describe the plant you are studying through other researchers. So this is the background knowledge that goes to the material chapter.
Response: Since the information indicated in the “results” section about the distribution of ―and common names for― concur with the findings of other research projects, to complement the description of the slender nightshade plants used in this study, other features relating to the harvesting area and the description of the harvested plant can be in the “methodology” section.
c) Line 345-346: "The mature fruits of divine nightshade were washed to remove the pulp, the seeds were extracted and dried at room temperature.". About what was the room temperature at that time? Describe in more detail the method of seed collection. When (date and year) were they collected? Was it a wild plant from which the seeds were collected? Were seeds collected from only one plant? Describe the place where the plant grows and give the region where it grew (coordinates or the exact village and county)? Has the author observed variability in wild species (e.g. were the seeds of the plant with the largest fruit collected)? Was a herbarium specimen also collected from the plant (if so, where herbarium specimen is it stored now?) or only the fruits? Are the seeds left over from the experiment stored somewhere, if so, in which institution and what is the deposit number?
Response: The methodology was supplemented in line with the indicated points.
d) Line 340-341: „The search included the keywords "divine nightshade," "hierba mora," "macuy," and "Solanum nigrescens" /---/“. But this plant has at least 26 Latin synonyms: https://powo.science.kew.org/taxon/urn:lsid:ipni.org:names:239356-2#synonyms In addition, in the English-language literature, this plant is called „Divine nightshade, Coastal-dune Nightshade, Slender nightshade“: https://identify.plantnet.org/the-plant-list/species/Solanum%20nigrescens%20M.%20Martens%20%26%20Galeotti/data . In Spanish, this plant is called "chiquiquelite, chichaquelite, mambia, hierba mora, maniloche, pitoxe": https://swbiodiversity.org/seinet/taxa/index.php?tid=3839&taxauthid=1&clid=2765# So the authors should better justify why they chose only a limited number of English and Spanish plant names for the search? They should also justify why Latin synonyms were not searched for when searching for information about the plant.
Response: The list of key search words was expanded and the information about Solanum nigrescens was checked.
e) The authors have thoroughly searched informations in several databases. Just for the authors to know in the future that older literature can also be searched in such databases: https://www.biodiversitylibrary.org/ ; https://bibdigital.rjb.csic.es/en/ . However, the information contained in the books can be searched in this database: https://books.google.com/ . The reviewer saw that there is quite a lot of information about this plant in the books.google.com databases. Of course, the authors have already covered most of the data on this topic in the manuscript. Therefore, it is not necessary to perform a new search in the databases. However, I also recommend using the book "Ethnobotany of Mexico" (2016) (pages 186 and 301 talk about this plant), which has not been used by the authors.
Response: The suggested databases were reviewed and appended to the information reported in the document.
f) Figure 10. The terms in the explanation of the figure should be changed. For example, instead of "open flowers" you could use "full-blown"; "fertilized blossom" could be used instead of "dead flowers"; instead of "fruit set" it could be either "developing fruits" or "ripening fruits"
Response: The indicated changes were made to the description of the figure.
g) Line 355. “The description included photographs of the most relevant stages.” Please write who took the plant photos in this manuscript? What month and year were the photos taken? Were the photos also processed or manipulated somehow (with what program?)? Please also indicate which camera the photos were taken with.
Response: The methodology pertaining to the description of the taking of photographs was supplemented, indicating the name of the photographer, the types of camera used, the year when the photos were taken, and the phorto-editing programs.
lines 55-57- please give more details for the difference between the two species (it should not only be the common "common name" Please provide morphological characters that are used as identification/diagnostic features because S. nigrum and S. nigrescens look the same at first glance and on pictures too. Highlight the differences besides the biological type (one is perennial and the other is annual). A suggestion - could be a small table comparing the leaves, indumentum, flowers fruits, and ranges of distribution.
Response: The information about the botanical and growth-habit differences between Solanum nigrecens and S. nigrum. was expanded, along with the toxicity reports for S. nigrum.

Reviewer 2 Report
The present paper deals with both 1) a review of scientific literature regarding the distribution, botanical characteristics, and uses, and 2) experimental tests on divine nightshade plants under greenhouse conditions namely their phenological development.
The paper is scientifically sound and well-written. It is quite well illustrated. However, there are two important points that must not be omitted.
First one is
lines 55-57- please give more details for the difference between the two species (it should not only be the common "common name" Please provide morphological characters that are used as identification/diagnostic features because S. nigrum and S. nigrescens look the same at first glance and on pictures too. Highlight the differences besides the biological type (one is perennial and the other is annual). A suggestion - could be a small table comparing the leaves, indumentum, flowers fruits, and ranges of distribution.
The second one is
Divine nightshade (Solanum nigrescens Mart. & Gal.) is a perennial, woody, and shrubby plant [10]. The common name is related to the European species (Solanum nigrum L.). In Mexico, it is also known as “Hierba de piojito”, rabbit weed, and lion's ear [11].
lines 359- 361
Further research, regarding its production under controlled conditions and the study of bioactive compounds present in the leaves and fruit, can lead to new high-nutritional and functional food products.
Since the plant is recommended for food in this paper it is absolutely necessary to add some detailed text about the toxicity - either in the introduction or in the results part. In my opinion, it would be better to add to the results part review of the toxicity. Its European relative Solanum nigrum is toxic for example.
Technical notes
In addition, the authors' names of the scientific name should not be repeated all the time. Once in the title and once in the text, (line 49), and once in the Abstract (line 8) is more than enough. Please delete the others.
Author Response
Response to Reviewer 2 Comments
Divine nightshade (Solanum nigrescens Mart. & Gal.) is a perennial, woody, and shrubby plant [10]. The common name is related to the European species (Solanum nigrum L.). In Mexico, it is also known as “Hierba de piojito”, rabbit weed, and lion's ear [11].
lines 359- 361
Further research, regarding its production under controlled conditions and the study of bioactive compounds present in the leaves and fruit, can lead to new high-nutritional and functional food products.
Since the plant is recommended for food in this paper it is absolutely necessary to add some detailed text about the toxicity - either in the introduction or in the results part. In my opinion, it would be better to add to the results part review of the toxicity. Its European relative Solanum nigrum is toxic for example.
Response: information about the toxicity of S. nigrum was added.
Technical notes
In addition, the authors' names of the scientific name should not be repeated all the time. Once in the title and once in the text, (line 49), and once in the Abstract (line 8) is more than enough. Please delete the others.
Response: The indications about observation techniques were heeded.

Round 2
Reviewer 1 Report
The authors have made the required revisions to the manuscript. I wish the authors success.